# Hepatic Glucose Metabolism Disorder Induced by Adipose Tissue-Derived miR-548ag via DPP4 Upregulation

**DOI:** 10.3390/ijms24032964

**Published:** 2023-02-03

**Authors:** Xiaolong Chu, Yanting Hou, Xueting Zhang, Menghuan Li, Dingling Ma, Yihan Tang, Chenggang Yuan, Chaoyue Sun, Maodi Liang, Jie Liu, Qianqian Wei, Yongsheng Chang, Cuizhe Wang, Jun Zhang

**Affiliations:** 1Medical College, Shihezi University, Shihezi 832000, China; 2Department of Medical Genetics, Medical College of Tarim University, Alaer 843300, China; 3Laboratory of Xinjiang Endemic and Ethic Diseases, Shihezi University, Shihezi 832000, China; 4Department of Physiology and Pathophysiology, Tianjin Medical University, Tianjin 300070, China

**Keywords:** obesity, miR-548ag, DNMT3B, DPP4, T2DM

## Abstract

The present study aimed to explore the molecular mechanism underlying the regulation of glucose metabolism by miR-548ag. For the first time, we found that miR-548ag expression was elevated in the abdominal adipose tissue and serum of subjects with obesity and type 2 diabetes mellitus (T2DM). The conditional knockout of adipose tissue *Dicer* notably reduced the expression and content of miR-548ag in mouse adipose tissue, serum, and liver tissue. The combined use of RNAseq, an miRNA target gene prediction software, and the dual luciferase reporter assay confirmed that miR-548ag exerts a targeted regulatory effect on DNMT3B and DPP4. miR-548ag and DPP4 expression was increased in the adipose tissue, serum, and liver tissue of diet-induced obese mice, while DNMT3B expression was decreased. It was subsequently confirmed both in vitro and in vivo that adipose tissue-derived miR-548ag impaired glucose tolerance and insulin sensitivity by inhibiting DNMT3B and upregulating DPP4. Moreover, miR-548ag inhibitors significantly improved the adverse metabolic phenotype in both obese mice and db/db mice. These results revealed that the expression of the adipose tissue-derived miR-548ag increased in obese subjects, and that this could upregulate the expression of DPP4 by targeting DNMT3B, ultimately leading to glucose metabolism disorder. Therefore, miR-548ag could be utilized as a potential target in the treatment of T2DM.

## 1. Introduction

Type 2 diabetes mellitus (T2DM) is a chronic metabolic disease characterized by elevated blood glucose levels and insulin resistance (IR). Obesity increases the risk of developing T2DM, although the specific molecular mechanism underlying this effect remains unclear.

Adipose tissue has high metabolic activity, and its core function is to maintain energy homeostasis through the regulation of adipokine release [1,2]. Adipokines released from adipose tissue act on other target cells and affect various physiological functions of the body, including glucose and fat metabolisms [3,4,5,6,7,8]. A novel kind of adipokine, miRNAs are released into circulation and transported to target tissues and organs where they participate in the regulation of various physiological processes. Aberrant expression and release of miRNAs may lead to diseases such as diabetes, tumors, and cardiovascular diseases [9,10,11,12,13,14,15]. Research on the role of miRNAs in glucose metabolism has increased, resulting in an expansion of the application prospects of miRNAs as diagnostic markers and therapeutic targets in the treatment of obesity-related metabolic diseases.

In a previous study conducted by our research group, the use of single nucleotide polymorphic chip screening revealed that miR-548ag exhibited a markedly positive correlation with fasting blood glucose (FPG) in 1053 subjects. No study so far has, however, demonstrated the biological function of miR-548ag in vivo. The miR-548 family is reported to have over 1000 members [14]. Most of the information on miR-548 family members concerns tumorigenesis [16,17], while knowledge regarding the relationship of miR-548 family members with obesity and T2DM remains limited. Zheng Yang et al. reported that miR-548 might be related to obesity-induced inflammatory responses, although the biological function of miR-548ag has not been reported to date. In the present study, SNP microarray chip technology was adopted to confirm, for the first time, that miR-548ag is positively correlated with blood glucose levels in study subjects. In addition, both the in vitro and in vivo experiments of the present study confirmed that increased miR-548ag levels are associated with obesity-induced abnormalities related to glucose metabolism in the body.

In the present study, the expression of miR-548ag in the abdominal adipose tissue of obese subjects was observed to be significantly higher than that in normal weight individuals. The conditional knockout of adipose tissue *Dicer* notably reduced the expression and content of miR-548ag in mouse adipose tissue, serum, and liver tissue. A bioinformatics analysis predicted that Dipeptidyl peptidase-4 (*DPP4*), which is related to glucose metabolism, could be the downstream target gene of miR-548ag. Previous studies have demonstrated that DPP4 is expressed highly in the liver and functions as a serine protease [18,19]. The aberrant expression of DPP4 is significantly related to the occurrence of obesity, diabetes, cancer, and other diseases. Therefore, DPP4 inhibitors are used widely in the treatment of T2DM [20,21].

The present study revealed that the expression of DPP4 was prominently upregulated after the overexpression of miR-548ag in both HepG2 and L02 cells. This is inconsistent with the classical pattern of the miRNA-based regulation of gene expression reported in the literature. The cause underlying the positive regulation of the expression of DPP4 by miR-548ag remains an intriguing scientific query that deserves further exploration.

The *DPP4* gene promoter region contains a huge number of CpG sites, which suggests that the expression of DPP4 may be regulated via methylation. The RNA sequencing results of the present study demonstrated that the expression of DNA methyltransferase 3B (DNMT3B) was downregulated after the upregulation of miR-548ag. In addition, a bioinformatics analysis predicted that the 3’UTR region of the DNMT3B mRNA contained a binding site for miR-548ag. Further analysis revealed that the increased levels of adipose tissue-derived miR-548ag after the onset of obesity upregulated the expression of DPP4 by suppressing DNMT3B in the liver, leading to abnormal glucose tolerance and decreased insulin sensitivity. Moreover, intraperitoneal injection of miR-548ag inhibitor could significantly improve glucose tolerance and insulin sensitivity in db/db mice via the downregulation of DPP4 expression in the liver.

## 2. Results

### 2.1. Adipose-Derived miR-548ag Levels Increased Conspicuously after the Onset of Obesity

The obese mouse model was successfully constructed after feeding the mice with 60% fat feed for 12 weeks. The body weight and Lee’s index of the mice were significantly increased in the obese mice, and the weight of the liver and adipose tissues were observably higher than those in the normal diet (ND) group (Figure 1A–D, Appendix A). Furthermore, the levels of GLU, FFA, TG, TC, HDL, and LDL were significantly higher in the obese mice (Figure 1E).

The levels of miR-548ag were significantly increased in the EpiWAT, serum, and liver tissue of the obese mice (Figure 1F). In addition, the serum miR-548ag levels in the subjects with obesity and T2DM were markedly increased compared with the normal weight (NC) individuals (Figure 1G). The miR-548ag levels in the abdominal adipose tissue of overweight and obese individuals were significantly higher than those in the NC group individuals (Figure 1H). In addition, the levels of serum miR-548ag were positively correlated with BMI, FPG, TG, TC, LDL, and HDL (Appendix A).

After the conditional knockout of adipose tissue *Dicer* and 12 weeks of feeding on 60% fat feed, the body weight, Lee’s index, and blood glucose levels in these mice were similar to those in the wild-type mice (Figure 1I–L). The expressions of *Dicer* and miR-548ag were dramatically reduced in the EpiWAT, and the miR-548ag content in the serum and the liver tissue was considerably decreased (Figure 1M,N).

### 2.2. DNMT3B and DPP4 Could Be the Key Target Genes of miR-548ag

The key downstream genes of miR-548ag were predicted using three databases (DIANA-MicroT, TargetScan, and miRWalk), and a total of 41 key genes were obtained by considering their intersection (Figure 2A). Subsequently, these 41 genes were functionally annotated using the DAVID database and the genetic association database (GAD-disease-class), which revealed 16 genes associated with metabolism (Appendix A). It was revealed thatDPP4 plays an important role in glucose metabolism [22,23,24]. After an intraperitoneal injection of miR-548ag, the expression levels of DPP4 in the liver tissue of the normal diet mice (ND) were correspondingly upregulated, which suggested the involvement of an intermediate molecule in the regulation of DPP4 expression by miR-548ag. Using the Ensemble database, it was determined that the GC content in the promoter region of *DPP4* was 44.1%, which suggested that this gene could be regulated via methylation. Subsequently, miR-548ag was upregulated in HepG2 cells, which were then subjected to transcriptome sequencing. The results revealed that *DNMT3B* expression was significantly reduced in these cells (Figure 2B). Intriguingly, an analysis performed using the TargetScan database identified miR-548ag-binding sites in *DNMT3B* (Figure 2C).

Next, to verify the targeted regulatory relationship between miR-548ag and *DNMT3B*, a miR-548ag mimic and the luciferase reporter plasmid of DNMT3B were transfected simultaneously into HepG2 cells. The results revealed that the miR-548ag mimic could markedly inhibit the luciferase activity in the 3’UTR region of DNMT3B-WT, although no noteworthy effect on DNMT3B-MUT was observed (Figure 2D,E). In addition, with the increase in the miR-548ag content in the adipose tissue, the serum, and the liver tissue of high-fat diet (HFD) mice (Figure 1F) compared with that of the ND group, the DNMT3B protein expression in the liver was significantly reduced while the DPP4 expression was increased (Figure 2F,G).

### 2.3. miR-548ag Impaired the Glucose Metabolism in Hepatocytes via DNMT3B Downregulation to Promote DPP4 Expression

After the successful overexpression of miR-548ag in HepG2 and L02 cells (Figure 3A), the protein expression of the DNMT3B was observed to be markedly decreased, while the expression of the DPP4 was increased (Figure 3B–E). In addition, miR-548ag overexpression significantly suppressed glucose consumption and insulin sensitivity (Figure 3F–H). After the transfection of the miR-548ag inhibitor into HepG2 and L02 cells, the protein expression of the DNMT3B was notably increased, while that of the DPP4 was decreased (Figure 3I–L). In addition, the glucose consumption and insulin sensitivity of the cells were observably increased (Figure 3M–O).

After the overexpression of DNMT3B in HepG2 and L02 cells (Figure 4A), the protein expression of DPP4 was significantly reduced (Figure 4B–E). In contrast, after the downregulation of DNMT3B (Figure 4F), the protein expression of DPP4 was significantly increased (Figure 4G–J). The overexpression of miR-548ag combined with a concomitant upregulation of DNMT3B significantly reversed the upregulation of DPP4 expression by the miR-548ag (Figure 4K–N) and also the inhibitory effects on cellular glucose consumption and insulin sensitivity (Figure 4O–R).

### 2.4. miR-548ag Impaired the Glucose Metabolism via DNMT3B Downregulation and DPP4 Upregulation In Vivo

The C57BL/6 male mice fed with the normal diet were intraperitoneally injected with adenovirus particles encoding the miR-548ag mimic once per week continuously for six weeks (Figure 5A). The body weight and the weights of the liver and adipose tissues increased remarkably (Figure 5B,C). An increase in miR-548ag expression was observed in the serum, liver tissue, and epididymal adipose tissue (Figure 5D). Moreover, the protein expression level of the DNMT3B was decreased in the liver, while the protein expression of level of the DPP4 was significantly increased (Figure 5E,F). The overexpression of miR-548ag impaired glucose tolerance and insulin sensitivity (Figure 5H–K). Furthermore, the GLU and the TC, TG, FFA, and LDL serum levels were prominently increased with the overexpression of miR-548ag (Figure 5G).

In order to explore whether a miR-548ag inhibitor could ameliorate glucose metabolism in HFD-induced obese mice, adenovirus particles encoding miR-548ag inhibitors were injected intraperitoneally into the mice once per week continuously for six weeks (Figure 6A). The miR-548ag inhibitor was able to significantly decrease the gain in body weight and the weights of the liver and adipose tissues induced by the HFD (Figure 6B,C). The protein expression of the DNMT3B was observably increased in the liver compared with that in the ND group, while the protein expression of the DPP4 was decreased (Figure 6D,E). The miR-548ag inhibitor improved glucose tolerance and insulin sensitivity in the HFD-induced obese mice (Figure 6G–J). Moreover, the levels of GLU, TC, TG, and LDL decreased significantly with the inhibition of miR-548ag (Figure 6F).

### 2.5. miR-548ag and DNMT3B Exerted no Significant Effect on the Methylation of the CpG Sites in the Promoter Region of DPP4

The results of bisulfite sequencing revealed that miR-548ag overexpression had no significant effect on the methylation state of the CpG sites in the promoter region of the *DPP4* in HepG2 cells (0 vs. 1.2%; Figure 7A; Appendix A). The overexpression of DNMT3B and the inhibition of DNMT3B also failed to alter the methylation state of the CpG sites in the promoter region of the *DPP4* (3.5% vs. 0 and 0 vs. 1.8%, respectively) in HepG2 cells (Figure 7B,C; Appendix A).

The overexpression of miR-548ag exerted no prominent effect on the methylation state of the CpG sites in the promoter region of the *DPP4* (0.5% vs. 0.5%) in the liver of ND group mice (Figure 7D; Appendix A). Additionally, the inhibition of miR-548ag exerted no significant effect on the methylation state of the CpG sites in the promoter region of the *DPP4* (0.5% vs. 0.3%) in the liver of the HFD-induced obese mice (Figure 7E; Appendix A).

### 2.6. Inhibition of miR-548ag Ameliorated Glucose Tolerance and Insulin Sensitivity via DNMT3B Upregulation and DPP4 Downregulation in db/db Mice

In order to investigate whether miR-548ag inhibitors could improve metabolic homeostasis, db/db mice were intraperitoneally injected with adenovirus particles encoding the miR-548ag inhibitor once per week continuously for six weeks (Figure 8A). In comparison with the control group, the injected db/db mice exhibited reduced body weight, liver and adipose tissue weights, and Lee’s index values (Figure 8B–D). Notably, the protein expression level of the DNMT3B was markedly increased in the liver of the mice with the miR-548ag inhibitor, while that of DPP4 was strongly suppressed (Figure 8E,F). The inhibition of miR-548ag improved glucose tolerance and insulin sensitivity. Moreover, the blood glucose levels were significantly decreased (Figure 8G–K) and the inhibition of miR-548ag ameliorated the lipid metabolism in the db/db mice (Figure 8G).

## 3. Discussion

Previous studies have demonstrated that adipose tissue is the main source of circulating miRNAs in vivo [25,26,27]. The present study has revealed that the content of miR-548ag in the abdominal adipose tissue of obese subjects was significantly higher than that of normal-weight subjects. In addition, it was demonstrated that with the increase in the body weight and fat content in the mice belonging to the high-fat diet group, the content of miR-548ag in the serum of the mice also increased significantly compared with that of mice in the normal diet group. Furthermore, in the high-fat diet group, the content of miR-548ag in the serum of the mice with the conditional knockout of adipose tissue *Dicer* was significantly reduced compared with that of the wild-type mice. Collectively, these data further corroborate the hypothesis that adipose tissue could be an important source of miR-548ag in vivo.

In the present study, the bioinformatics analysis predicted that *DPP4* could be the downstream target gene of miR-548ag. The expression of DPP4 in the liver was observed to be significantly higher than that in the other tissues and organs [28]. Recent studies have demonstrated that DPP4 inhibitors that directly target the liver could be effective in relieving IR, which could be a better and more effective strategy for treating diabetes [29]. Previous studies have demonstrated that miRNAs bind to the 3’UTRs of their target genes, resulting in the termination of translation or the direct degradation of the target mRNAs [30]. Intriguingly, in the present study, miR-548ag promoted the expression of its downstream target gene *DPP4*, although the specific molecular mechanism underlying this effect warrants further investigation.

DNA methylation is one of the major epigenetic mechanisms affecting gene expression and genome stability [31,32]. Recent studies have demonstrated that the disruption of DNA methylation, histone modifications, and RNA-mediated biological homeostasis due to obesity may lead to multiple pathological changes, ultimately causing T2DM [33]. In the present study, the Ensembl database analysis revealed that the CG content in the promoter region of *DPP4* was greater than 40%. However, whether *DPP4* expression is regulated via methylation cannot be reported. In the present study, in both HepG2 and L02 cells, the upregulation and the downregulation of DNMT3B led to a decrease and increase, respectively, in the expression of intracellular DPP4. Moreover, the dual luciferase reporter assay confirmed that miR-548ag can bind to the 3’UTR of DNMT3B. In addition, both the in vitro and in vivo experiments verified that miR-548ag exerts a negative regulatory effect on the expression of DNMT3B in the liver tissues/cells. These results suggest that miR-548ag could promote the expression of DPP4 in hepatocytes via targeted downregulation of DNMT3B after the onset of obesity, which could eventually lead to glucose metabolism disorder. Whether DNMT3B regulates the expression of DPP4 by directly affecting the methylation status of the CpG sites in the promoter region of *DPP4* remains to be confirmed by further investigation.

The subsequent in vitro and in vivo experiments revealed that, after the upregulation of miR-548ag and the up/down-regulation of DNMT3B, although the protein expression level of the DPP4 was correspondingly up/down-regulated, the methylation status of the CpG sites in the promoter region of the *DPP4* remained unaffected. The above results verify that DNMT3B exerts a significant effect on the expression of DPP4, although this effect is not achieved through a direct alteration of the methylation status of the CpG sites in the promoter region of the *DPP4*, and other molecular mechanisms must be involved. Using the PROMO database and the methyl primer design software, it was subsequently revealed that 12 of the 21 potential *DPP4* transcription factors contained CpG islands (YY1, CEBPB, XBP1, TBP, ETS1, PAX5, TP53, MYC, GCFC2, FOS, JUN, and NF1) [34,35,36,37,38,39,40] (Appendix A). The question of whether the effect of DNMT3B on the expression of DPP4 is achieved through the above transcription factors is worthy of further investigation.

Currently, DPP4 inhibitors are used widely in the treatment of T2DM [20,21]. However, recent studies have revealed certain major side effects of DPP4 inhibitors, including a significant increase in the risk of heart failure, death, myocardial infarction, stroke, and a cumulative gout incidence of 5.25% [41,42]. In addition, DPP4 inhibitors may cause infectious diseases, thereby greatly increasing the risk of venous thromboembolism [43]. Numerous studies have indicated that miRNAs regulate insulin sensitivity, and this effect may be utilized in the treatment of obesity and T2DM [44]. The present study revealed that an miR-548ag inhibitor could significantly reduce the body weight and the weights of the liver and adipose tissues in db/db mice. Moreover, glucose tolerance and insulin sensitivity were also significantly improved in the db/db mice. The above results suggest that miR-548ag could be utilized as a potential target in the treatment of obesity-induced T2DM.

The adipose tissue-derived miR-548ag promoted the expression of DPP4 by targeting the downregulation of DNMT3B in the liver, which ultimately caused abnormal glucose tolerance and decreased insulin sensitivity. Furthermore, it was confirmed that the effect of DNMT3B on DPP4 expression was not achieved through the direct alteration of the methylation status of the CpG sites in the promoter region of the *DPP4*. Accordingly, it is suggested that miR-548ag could be used as a potential target in the treatment of obesity-induced T2DM.

## 4. Material and Methods

### 4.1. Human Samples

The serum samples of 220 individuals from the Xinjiang province of China were collected between the years 2019 and 2020. General information on body weight, WC, BMI, FPG, and plasma levels of TG, TC, HDL, and LDL was included. The subjects were divided into three groups: a normal-weight group (NC, *n* = 61, 18.5 kg/m^2^ ≤ BMI ≤ 24 kg/m^2^), an obese group (Ob, *n* = 103, BMI ≥ 28 kg/m^2^), and a type 2 diabetes group (T2DM, *n* = 56, FPG ≥ 7.0 mmol/L, 2 h PG ≥ 11.1 mmol/L). The abdominal adipose tissue samples of 21 subjects were collected between May 2021 and June 2022 at the First Affiliated Hospital, Shihezi University School of Medicine. The samples included 11 normal-weight subjects and 10 overweight/obese subjects.

T2DM diagnosis was established based on the diagnostic criteria of the World Health Organization (issued in 1999) and the Guidelines for the Prevention and Treatment of Type 2 Diabetes Mellitus in China (2020 edition). Exclusion criteria were as follows: type 1 diabetes, tumors, acute inflammation, kidney disease, and recent consumption of drugs known to interfere with glucose and lipid metabolisms. All subjects provided informed and voluntary consent prior to enrollment in the present study, and it was understood by the subjects that the clinical information and biological samples would be used for research and publication. The consent form and the ethical approval for conducting the study were provided by the Medical Ethics Committee of the First Affiliated Hospital, Shihezi University School of Medicine (reference number 2019–029–01).

### 4.2. Measurement of Biochemical Indices

The weights, waist measurements, glycemic index values, and BMIs of all included individuals were obtained. The levels of FPG, TC, TG, HDL, and LDL were detected using an automated biochemistry analyzer (Mindray, USA).

### 4.3. Animal Care

C57BL/6 male mice were procured from HUNAN SJA Laboratory Animal Co., Ltd. (Hunan, China). *Lepr*^db/db^ and *Lepr*^db/−^ male mice were procured from the Model Animal Research Center, Changzhou Caverns (Changzhou, China). The *Dicer*^−/−^ mice were procured from Cyagen Biosciences (Guangzhou, China). All mice were housed in groups of five animals per cage in an SPF facility with the following conditions: a temperature of 22–24 °C, humidity of 40–50%, and a 12-h light/dark cycle. All animal care and handling procedures were performed according to international laws and policies. The animal experiments used in the present study were approved by the animal ethics committee of the First Affiliated Hospital, Shihezi University (A2019–087–01).

### 4.4. Cell Culture

HepG2 and L02 cells were purchased from the Cell Bank of the Chinese Academy of Sciences (Shanghai, China) and cultured in 25 mmol/L glucose Dulbecco’s modified Eagle’s medium (DMEM, Gbico) supplemented with 10% fetal bovine serum and 1% penicillin/streptomycin (100 µg/mL). The cells were cultured at 37 °C inside an incubator with a humidified atmosphere containing 5% CO_2_. After transfection, the RNA and protein were extracted from the cells, or an insulin sensitivity assay was performed. The 293A cells were purchased from Procell (Wuhan, China) and cultured in 25 mmol/L glucose Dulbecco’s modified Eagle’s medium (DMEM, Gbico) supplemented with 10% fetal bovine serum and 1% penicillin/streptomycin (100 µg/mL). These cells were also cultured at 37 °C in a humidified atmosphere containing 5% CO_2_.

### 4.5. Adenovirus Amplification and Purification

The miRNA-548ag mimic adenovirus vector and inhibitor used in the present study were constructed at and provided by Shanghai GenePharma. The adenovirus was amplified in 293A cells. In brief, the 293A cells were cultured in a medium supplemented with 10% fetal bovine serum, and when approximately 90% confluence was reached, the adenovirus seed was added to these cells for transfection. After 48 h of transfection, visible regions of cytopathic effect (CPE) were observed. The adenovirus stock was harvested when approximately 90% CPE was observed (typically 72 h post-transfection). The cell suspension was freeze-thawed three times in a methanol–dry ice mix bath and a water bath at 37 °C. Afterward, the supernatant was collected through centrifugation and frozen at –80 °C for long-term preservation. The purification procedures were conducted according to the instructions of the ViraTrap “M Adenovirus Purification Maxiprep KitViraTrap” adenovirus mass purification kit, with the reagent dosage adjusted in proportion to the virus suspension quantity.

### 4.6. Intraperitoneal Injection of the Adenovirus Vector in Mice

Sixteen four-week-old C57BL/6 male mice were fed with a normal diet (D12450J, 10% energy from fat) until the sixteenth week. These mice were then divided into two groups: the mice that received an intraperitoneal injection of empty adenovirus (*n* = 8) and the mice that received an intraperitoneal injection of adenovirus particles encoding the miR-548ag mimic (*n* = 8, 1 × 10^11^ VP/mice) once per week for six weeks. A different set of sixteen four-week-old C57BL/6 male mice were fed a high-fat diet (D12494, 60% energy from fat) until the sixteenth week. These mice were then divided into two groups: the mice that received an intraperitoneal injection of empty adenovirus (*n* = 8) and the mice that received an intraperitoneal injection of the adenovirus particles containing the miR-548ag-inhibitor (*n* = 8, 1 × 10^11^ VP/mice) once per week for six weeks. Another set of twelve four-week-old C57BL/6 db/db male mice was subjected to adaptive feeding for one week, after which these mice were divided into two groups: the mice that received an intraperitoneal injection of empty adenovirus (*n* = 6) and the mice that received an intraperitoneal injection of adenovirus particles containing the miR-548ag inhibitor (*n* = 6, 1 × 10^11^ VP/mice) once per week for six weeks.

### 4.7. Cell Transfection

The miR-548ag mimic, the miR-548ag inhibitor, and the DNMT3B interference fragment were transfected into cells using Lipofectamine 2000 (Catalog#: 11668–019; Invitrogen, Waltham, MA, USA), while Lipofectamine 3000 was used when transfecting the DNMT3B-overexpression plasmid. HepG2 cells were transfected with the miR-548ag mimic, miR-548ag inhibitor, DNMT3B interference fragment, and DNMT3B-overexpression plasmid at respective concentrations of 50 nM, 100 nM, 80 nM, and 4 µg/mL. L02 cells were transfected with the miR-548ag mimic, miR-548ag inhibitor, DNMT3B interference fragment, and DNMT3B-overexpression plasmid at respective concentrations of 50 nM, 100 nM, 40 nM, and 1 µg/mL. After 4–6 h of transfection, the transfection reagent was replaced with DMEM supplemented with 10% fetal bovine serum. After 24 h, the cells were subjected to RNA or protein extraction or other treatments. The miR-548ag mimic, the miR-548ag inhibitor, and the DNMT3B interference fragments used in this experiment were synthesized at and provided by Shanghai GenePharma (Shanghai, China). The DNMT3B-overexpression plasmid was constructed at Beijing Hesheng Biotechnology (Beijing, China).

### 4.8. Cellular Glucose Consumption and Insulin Sensitivity Assay

Cellular glucose consumption assay: After 12 h of serum-free starvation, the cells were transfected with the mimic/inhibitor for 24 h, following which the culture medium was collected for glucose concentration measurements using the glucose oxidase method.

Cellular insulin sensitivity assay: After 12 h of serum-free starvation, the cells were transfected with the mimic/inhibitor for 24 h, following which the cells were stimulated continuously with a 100 nmol/L insulin + low-glucose DMEM (1 g/L) + 10% FBS mix medium. The culture medium was collected at 0 min, 15 min, 30 min, 45 min, 60 min, 90 min, and 120 min for glucose concentration measurements.

### 4.9. Western Blotting and Antibodies

The total protein from the liver, HepG2 cells, and L02 cells was extracted using an RIPA Lysis Buffer (Cat#R0010, Solarbio) containing 1% PMSF (Cat# P8340, SolarBio). The lysates were quantified, following which equal amounts of protein were subjected to SDS-PAGE and immunoblotted with antibodies against GAPDH, DNMT3B, and DPP4. The antibodies against GAPDH (TA-08) were obtained from ZSGB-BIO, and the antibodies against DNMT3B (Ab79822) and DPP4 (Ab129060) were purchased from Abcam.

### 4.10. Real-Time PCR

A miRcute Plus microRNA first-strand cDNA kit and a miRcute plus microRNA qPCR Kit were purchased from TIANGEN (Beijing, China). A total mRNA reverse transcription kit was purchased from ThermoFisher Scientific (Waltham, MA, USA). A real-time quantitative PCR kit was purchased from QIAGEN (Hilden, Germany). All PCR procedures were performed according to the instructions provided in the respective kits. The primers used in the PCR are provided in Appendix A.

### 4.11. Dual-Luciferase Reporter Assay

The TargetScan database (http://www.targetscan.org) was used for analyzing the 3’UTR binding sites of miR-548ag and DNMT3B. The Luciferase plasmids were constructed at and provided by Shanghai GenePharma: hDNMT3B-mir-548ag wt (5’-TTTAGGCTGAAAGATGACGGATGCCTAGAGTTTACCTTATGTTTAATTAAAATCAGTATTTGTCT-3’) and hDNMT3B-mir-548ag mut (5’-TTTAGGCTGAAAGATGACGGATGCCTAGAGTaatggaaATGTTTAATTAAAATCAGTATTTGTCT-3’). The dual luciferase activity assay was performed according to the manufacturer’s instructions (Promega).

### 4.12. Plasmid Amplification and Extraction

The bacteria were incubated in the LB medium (peptone: 10 g/L, yeast powder: 5 g/L, NaCl: 10 g/L). After adding the antibiotics, the bacteria were enriched in a constant temperature shaker for 12 h. The plasmid was extracted according to the instructions provided in the plasmid extraction kit (TIANGEN).

### 4.13. Intraperitoneal Glucose Tolerance and Insulin Sensitivity Evaluations in Mice

The glucose tolerance of the mice was evaluated after 12 h of overnight fasting. After determining the FBG levels, an intraperitoneal bolus of 2 g of glucose per kg of body weight was administered to each mouse. The blood glucose levels were then detected after 15 min, 30 min, 45 min, 60 min, and 120 min.

The insulin sensitivity of the mice was evaluated after 6 h of fasting. After determining the FBG levels, an intraperitoneal bolus of 0.5 UI insulin per kg of body weight was administered to each mouse. The blood glucose levels were detected after 15 min, 30 min, 45 min, 60 min, and 120 min.

### 4.14. Statistical Analysis

The SPSS statistical package (version 17.0, SPSS Inc, Chicago, IL, USA) was employed for the statistical analysis of the resulting data. For the normally distributed data, the statistical differences between the groups were determined using an unpaired Student’s *t*-test, while a rank-sum test was performed for the non-normally distributed data. A Chi-squared test was used for comparing the numerical data. *p* < 0.05 was used as the threshold of statistical significance.

## Figures and Tables

**Figure 1 ijms-24-02964-f001:**
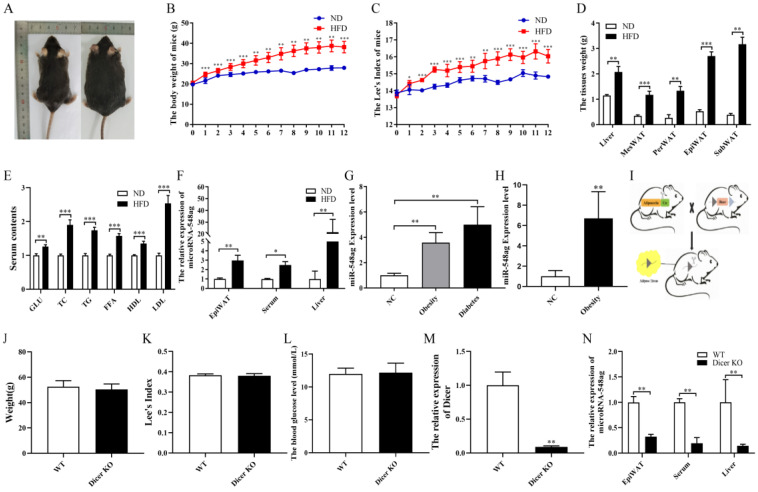
Effect of obesity on miR-548ag expression levels. (**A**) The gross morphology of the ND and HFD mice. (**B**) The body weight (*n* = 8). (**C**) Lee’s index (*n* = 8). (**D**) The weights of the liver, mesenteric adipose tissue (MesWAT), perirenal adipose tissue (PerWAT), epididymal adipose tissue (EpiWAT), and subcutaneous adipose tissue (SubWAT) (*n* = 8). (**E**) The levels of blood glucose, TC, TG, FFA, HDL, and LDL (*n* = 8). (**F**) The levels of miR-548ag in the serum, the liver, and the EpiWAT of the mice (*n* = 6). (**G**) The serum miR-548ag levels as determined from serum samples from the normal weight (NC, *n* = 61), obesity (*n* = 103), and diabetes (*n* = 56) groups. (**H**) The miR-548ag levels in the abdominal adipose tissue as determined from abdominal adipose tissue samples from the NC (*n* = 8) and obesity (*n* = 11) groups. (**I**) Construction of the adipose tissue *Dicer* knockout mouse model. (**J**–**L**) The body weight, Lee’s index, and fasting glucose levels in the adipose tissue *Dicer* knockout mice (*n* = 6). (**M**) The expression of *Dicer* in the adipose tissue (*n* = 6). (**N**) The expression of miR-548ag in the EpiWAT, serum, and liver tissue of the adipose tissue *Dicer* knockout mice (*n* = 6). (The *p*-values from the *t*-test and non-parametric rank sum test are indicated. The data are presented as mean ± SEM. * *p* < 0.05, ** *p* < 0.01, and *** *p* < 0.001 indicate a significant difference.)

**Figure 2 ijms-24-02964-f002:**
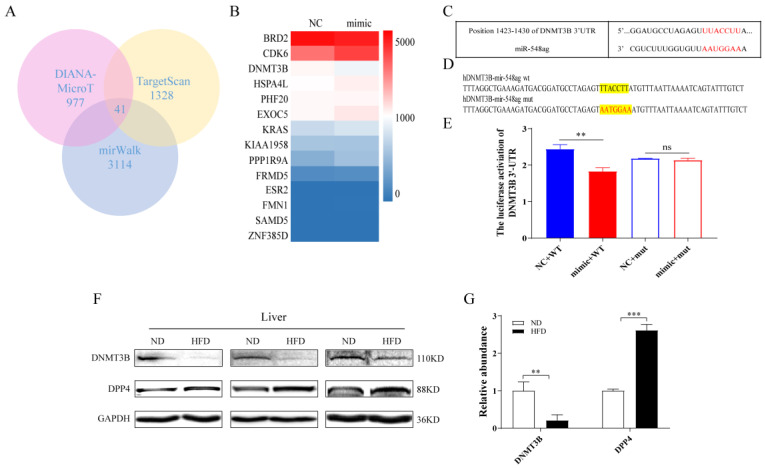
The prediction of the downstream target gene of miR-548ag. (**A**) The DIANA-MicroT, TargetScan, and miRWalk databases were employed to predict the key downstream genes of miR-548ag, and the intersection of the three databases was determined. (**B**) The transcriptome sequencing analysis of mir-548ag upregulated in HepG2 cells. (**C**) The TargetScan database analysis of the miR-548ag binding sites for DNMT3B. (**D**) Construction of a dual luciferase reporter gene plasmid for DNMT3B. (**E**) Analysis of the results of the dual luciferase reporter assay. (**F**,**G**) The protein expressions of DNMT3B and DPP4 in the liver. (The *p*-values obtained in the *t*-test and non-parametric rank sum test are indicated. The data are presented as mean ± SEM. ** *p* < 0.01, and *** *p* < 0.001 indicate a significant difference.)

**Figure 3 ijms-24-02964-f003:**
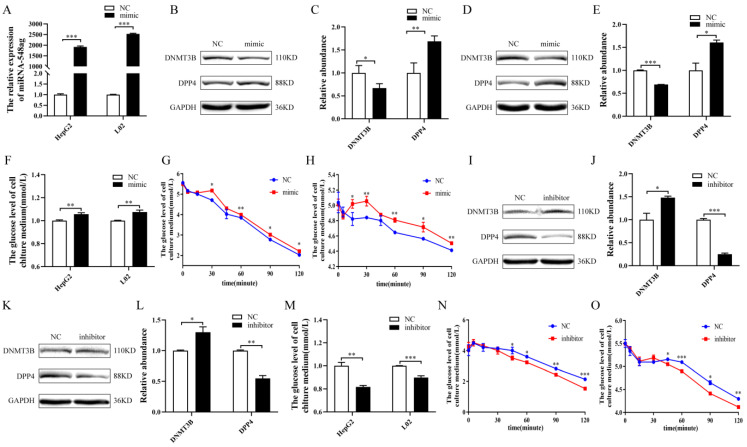
Effect of miR-548ag on the expressions of DNMT3B and DPP4. (**A**) The expression of miR-548ag in HepG2 and L02 cells was upregulated when 50 nM of miR-548ag mimic was used. (**B**–**E**) The protein expressions of DNMT3B and DPP4 in HepG2 and L02 cells after the overexpression of miR-548ag. (**F**) The glucose consumption, (**G**,**H**) insulin sensitivity, and (**I**–**L**) protein expressions of DNMT3B and DPP4 in HepG2 and L02 cells after transfection with 100 nM miR-548ag inhibitor for 24 h. (**M**) Glucose consumption after the inhibition of miR-548ag. (**N**,**O**) Insulin sensitivity after the inhibition of miR-548ag. (The *p*-values obtained in the *t*-test and non-parametric rank sum test are indicated. The data are presented as mean ± SEM. * *p* < 0.05, ** *p* < 0.01, and *** *p* < 0.001 indicate a significant difference).

**Figure 4 ijms-24-02964-f004:**
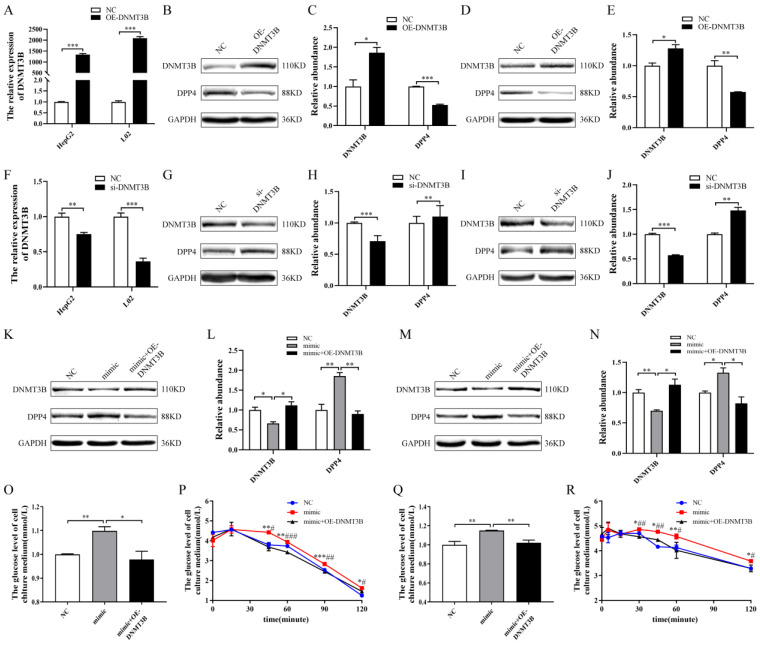
Rescue assay of HepG2 and L02 cells. (**A**) The mRNA expression of DNMT3B in HepG2 and L02 cells when 4 µg/mL and 1 µg/mL concentrations of DNMT3B-overexpression plasmid were used, respectively. (**B**–**E**) The protein expressions of DNMT3B and DPP4 in HepG2 and L02 cells after the overexpression of DNMT3B. (**F**) The mRNA expression of DNMT3B in HepG2 and L02 cells when 80 nM and 40 nM of the DNMT3B interference fragments were used, respectively. (**G**–**J**) The protein expressions of DNMT3B and DPP4 in HepG2 and L02 cells after the downregulation of DNMT3B. (**K**–**N**) The protein expressions of DNMT3B and DPP4 in HepG2 and L02 cells after overexpressing miR-548ag and upregulating DNMT3B simultaneously. (**O**,**P**) The glucose consumption and insulin sensitivity in HepG2 cells after overexpressing miR-548ag and upregulating DNMT3B simultaneously. (**Q**,**R**) The glucose consumption and insulin sensitivity in L02 cells after overexpressing miR-548ag and upregulating DNMT3B simultaneously. (The *p*-values obtained in the *t*-test and non-parametric rank sum test are indicated. The data are presented as mean ± SEM. * *p* < 0.05, ** *p* < 0.01, and *** *p* < 0.001 indicate a significant difference, Figure P and Figure R, mimic group vs. NC group; * *p* < 0.05, ** *p* < 0.01, and *** *p* < 0.001 indicate a significant difference, mimic group vs. mimic + ad-DNMT3B group; ^#^
*p* < 0.05, ^##^
*p* < 0.01, and ^###^
*p* < 0.001 indicate a significant difference).

**Figure 5 ijms-24-02964-f005:**
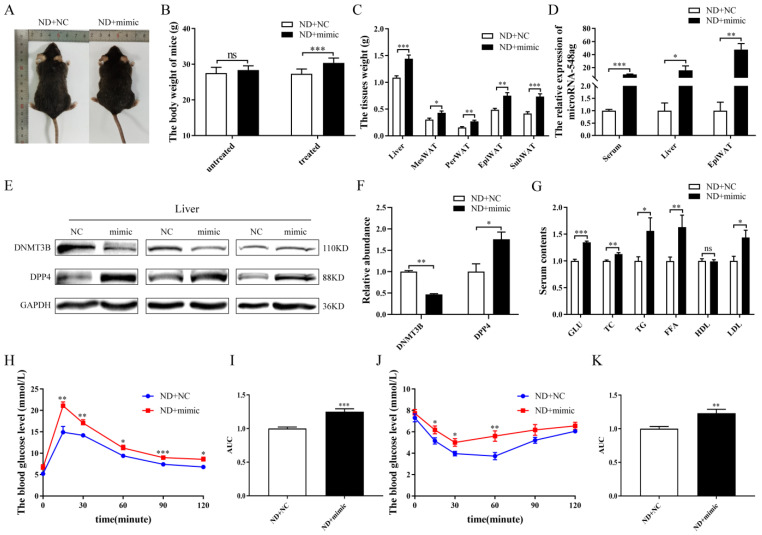
Effect of miR-548ag on the expression levels of DNMT3B and DPP4 in vivo. (**A**) The gross morphology of ND + NC and ND + mimic mice. (**B**) Comparison of body weights prior to and after the intraperitoneal injection of the mimic (*n* = 8). (**C**) The weights of the liver and adipose tissues (*n* = 8). (**D**) The levels of miR-548ag in the serum, liver tissue, and EpiWAT of the mice (*n* = 6). (**E**,**F**) The protein expressions of DNMT3B and DPP4 in the liver. (**G**) The levels of blood glucose, TC, TG, FFA, HDL, and LDL (*n* = 8). (**H**) IPGTT assay results (*n* = 8). (**I**) The quantification of glucose tolerance. (J) ITT assay results (*n* = 8). (**K**) The quantification of insulin sensitivity. (The *p*-values obtained in the *t*-test and non-parametric rank sum test are indicated. The data are presented as mean ± SEM. * *p* < 0.05, ** *p* < 0.01, and *** *p* < 0.001 indicate a significant difference).

**Figure 6 ijms-24-02964-f006:**
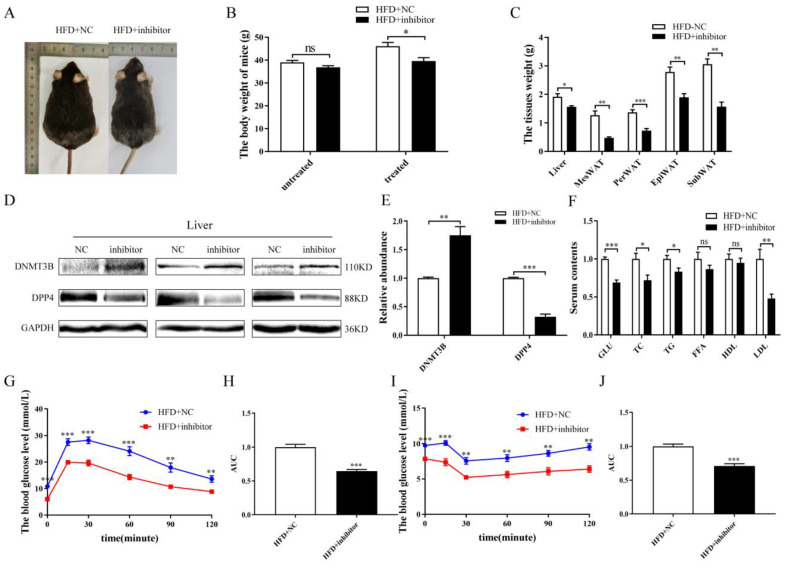
Effect of the miR-548ag inhibitor on the expressions of DNMT3B and DPP4 in vivo. (**A**) The gross morphology of the HFD + NC and HFD + inhibitor mice. (**B**) Comparison of the body weights prior to and after the intraperitoneal injection of the inhibitor (*n* = 8). (**C**) The weights of the liver and adipose tissues (*n* = 8). (**D**,**E**) The protein expressions of the DNMT3B and DPP4 in the liver (**F**) The levels of blood glucose, TC, TG, FFA, HDL, and LDL (*n* = 8). (**G**) IPGTT assay results (*n* = 8). (**H**) The quantification of glucose tolerance. (**I**) ITT assay results (*n* = 8). (**J**) The quantification of insulin sensitivity. (The *p*-values obtained in the *t*-test and non-parametric rank sum test are indicated. The data are presented as mean ± SEM. * *p* < 0.05, ** *p* < 0.01, and *** *p* < 0.001 indicate a significant difference).

**Figure 7 ijms-24-02964-f007:**
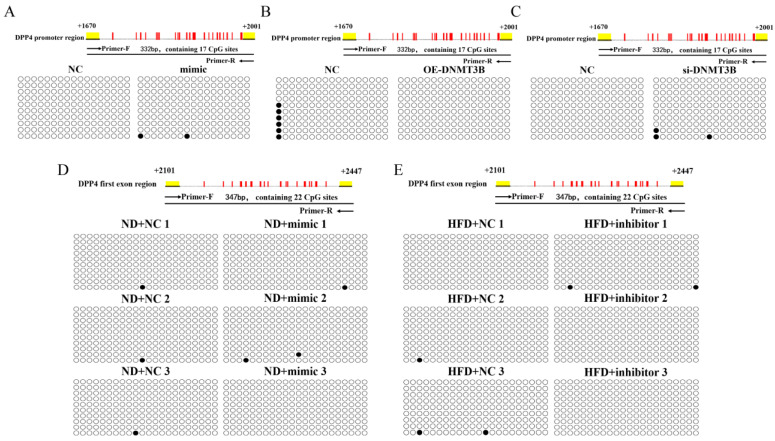
The BSP experiment. (**A**) Schematic diagram of the methylation status of the CpG sites in the promoter region of the *DPP4* gene. HepG2 cells were transfected with 50 nM miR-548ag mimic for 24 h after the upregulation of miR-548ag. (**B**,**C**) The schematic diagrams of the methylation status of the CpG sites in the promoter region of the *DPP4* gene. The transfection of 4 µg/mL of the DNMT3B-overexpression plasmid and 40 nM of the DNMT3B interference fragment into HepG2 cells was performed for 24 h after the up/down-regulation of DNMT3B. (**D**) The methylation status of the CpG sites in the promoter region of the mouse liver *DPP4* gene. Adenovirus particles encoding the miR-548ag mimic were injected intraperitoneally into the normal diet-fed mice for 6 weeks. (**E**) Schematic diagram of the methylation status of the CpG sites in the promoter region of the mouse liver *DPP4* gene. Adenovirus particles with the miR-548ag inhibitor were injected intraperitoneally into the HFD-induced obese mice for 6 weeks. ○ represents the non-methylated CpG site; ● represents the methylated CpG site.

**Figure 8 ijms-24-02964-f008:**
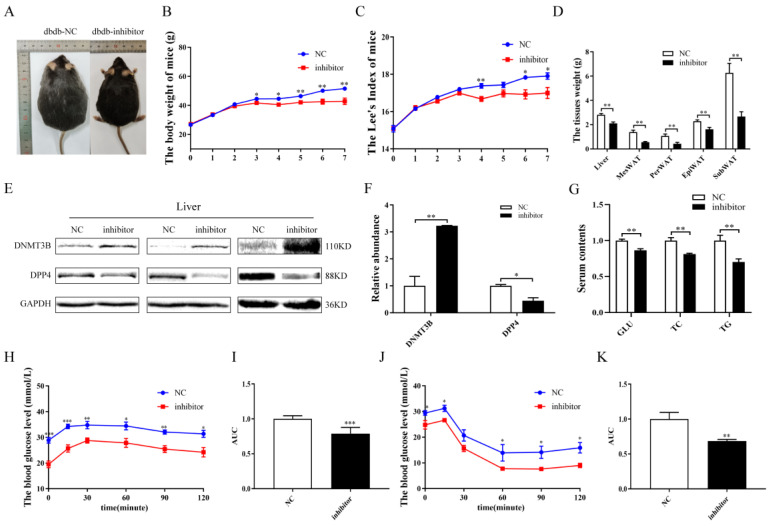
Effect of the miR-548ag inhibitor on db/db mice. (**A**) The gross morphology of the db/db + NC and db/db + inhibitor mice. (**B**) Comparison of the body weights prior to and after the intraperitoneal injection of the inhibitor (*n* = 6). (**C**) Lee’s index (*n* = 6). (**D**) The weight of the liver tissue, MesWAT, PerWAT, EpiWAT, and SubWAT (*n* = 6). (**E**,**F**) The protein expressions of DNMT3B and DPP4 in the liver. (**G**) The levels of blood glucose, TC, and TG (*n* = 6). (**H**) IPGTT assay results (*n* = 6). (**I**) The quantification of glucose tolerance. (**J**) ITT assay results (*n* = 6). (**K**) The quantification of insulin sensitivity. (The *p*-values obtained in the *t*-test and non-parametric rank sum test are indicated. The data are presented as mean ± SEM. * *p* < 0.05, ** *p* < 0.01, and *** *p* < 0.001 indicate a significant difference.)

## Data Availability

All data and materials related in this research are available for sharing.

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
