# Peer review of "Hepatic Glucose Metabolism Disorder Induced by Adipose Tissue-Derived miR-548ag via DPP4 Upregulation"

_ijms, 2023, doi:10.3390/ijms24032964_

Round 1

Reviewer 1 Report

1. At the beginning of the abstract there should rather not be information about previous research. the abstract is about this particular work
2.In the Introduction paragraph, the abbreviation DPP4 is not explained, it only appears in the Results paragraph.
3.There is too little information about the rationale that led the authors to study specifically DPP4.
4.The discussion needs to be rewritten. Some information, such as general information about mi-RNAs, as well as information about the miR-548 family, should rather be included in the Introduction paragraph.
5. mi-RNA is unlikely to be called an adipocytokine

Author Response

Submission ID: IJMS-2176846

Answer the question of Reviewer comments

感谢您给我们机会修改文章:脂肪组织来源的miR-548ag通过DPP4上调诱导肝糖代谢紊乱。审稿人提出了一些宝贵的意见。我们将对您提出的问题进行解释和回答,然后在文章的相应位置进行调制。

对作者的意见和建议

1.At 摘要的开头,就不应该有关于以前研究的信息。摘要是关于这个特殊的工作

答:感谢您的评论,正如您所说,太多与研究无关的内容会使摘要变得繁琐和不集中。因此,我们从修订稿的摘要中删除了以前的研究,这将使读者通过阅读摘要来快速了解本文的研究内容。再次感谢您的评论。

2.In 引言段落中,没有解释缩写DPP4,它只出现在结果段落中。

答:非常感谢您的仔细审查,由于我们的疏忽,缩写DPP4首次出现时没有解释,我们在介绍的第三段中首次出现的地方添加了DPP4的全名。

3.关于导致作者专门研究DPP4的基本原理的信息太少。

答:感谢您的宝贵意见。我们不准确的描述无法让您很好地理解为什么在这项研究中选择DPP4。对此,我们深表歉意。我们选择DPP4进行深入研究的原因如下:

(1)生物信息学分析预测,与糖代谢相关的二肽基肽酶-4(DPP4)可能是miR-548ag的下游靶基因。

(2)已有研究表明,DPP4的异常表达与肥胖和糖尿病的发生有显著相关性。此外,DPP4抑制剂广泛用于治疗T2DM[1,2]。

(3)本研究的实验前结果表明,miR548ag的上调显著促进了体外培养肝细胞中DPP4的表达。上述结果与经典miRNA调控机制不一致。这促使我们进一步研究miR548ag上调DPP4表达的具体分子机制。RNA测序结果表明,miR-548ag上调后DNMT3B的表达下调。此外,生物信息学分析预测DNMT3B mRNA的3'UTR区域含有miR-548ag的结合位点。DNA序列分析结果表明,DPP4基因启动子区含有大量的CpG位点(图1),这表明DPP4的表达可能通过甲基化调控。基于上述实验结果和现有文献,我们提出了miR548ag可以通过靶向DNMT3B表达上调DPP4的假设。

In summary, this study identified an intrinsic link between miR548ag-DNMT3B-DPP4. Therefore, we chose DPP4 for a deep study. Thank you again for your valuable feedback, we hope the above answers help you understand why we chose DPP4.

Figure 1 DNA sequence analysis of the promoter region of DPP4 gene with a large number of CpG sites

4. The discussion needs to be rewritten. Some information, such as general information about mi-RNA, and information about the miR-548 family, should be included in the introduction paragraph.

A: Thank you very much for your valuable comments, and we will correct the discussion section in the revised draft as follows:

(1) We have adjusted the general information about mi-RNAs, as well as information about the miR-548 family to the introduction according to your comments.

(2) Based on your comments, we have carefully reviewed the discussion section of this paper and found that the descriptions of both DPP4 and miR548ag are duplicated, so we have removed some duplicates in the revised manuscript.

(3) After careful evaluation of the discussion section at your suggestion, we also found that the discussion on the regulation of DPP4 expression by methylation does not make much logical sense. In the revised manuscript we have adjusted this section as follows: first discuss the effect of DNMT3B on DPP4 expression, and then elaborate on the targeted inhibitory effect of miR548ag on DNMT3B expression.

Thanks again for your valuable suggestions, which will make the discussion part of this article more reasonable and logical, so that readers can better understand our article and get some inspiration from it.

  1. mi-RNA is unlikely to be called an adipocytokine

Answer: Thank you for your careful review, your comments are very accurate, after reviewing the literature we found that calling miRNA as adipocytokines is not accurate and not strict enough, it should be called adipokine . We have replaced adipocytokines with adipokines in the revised manuscript and added the corresponding references[3]. Thank you again for your suggestion.

References

  1. Deacon, C.F. Dipeptidyl peptidase 4 inhibitors in the treatment of type 2 diabetes mellitus. Nat Rev Endocrinol. 2020,16,642–653.
  2. Sachinidis, A.; Nikolic, D.; Stoian, A. P.; Papanas, N.; Tarar, O.; Rizvi, A. A.; & Rizzo, M. Cardiovascular outcomes trials with incretin-based medications: a critical review of data available on GLP-1 receptor agonists and DPP-4 inhibitors. Metabolism. 2020,111,154343.
  3. Hong, P., Yu, M., & Tian, W.. Diverse RNAs in adipose-derived extracellular vesicles and their therapeutic potential. Molecular therapy. Nucleic acids, 202126, 665–677.

Reviewer 2 Report

In their study, Chu et al tried to investigate the potential link between miR-548ag and hepatic glucose metabolism via DPP4. The data generated seems to be robust and statistical analysis of the data appears rigorous. The majority of the discussion section is consistent with the study's results.

Minor points:

·       Line 73, T2DM Abbreviation should be predefined.

·       Authors are advised to include Aim/objective of the study.

·       Line 88-90, please add a reference.

·       Conclusion section is missing.

Author Response

提交编号: IJMS-2176846

回答审稿人评论的问题

感谢您给我们机会修改文章:脂肪组织来源的miR-548ag通过DPP4上调诱导肝糖代谢紊乱。审稿人提出了一些宝贵的意见。我们将对您提出的问题进行解释和回答,然后在文章的相应位置进行调制。

对作者的意见和建议

在他们的研究中,Chu等人试图通过DPP4研究miR-548ag与肝糖代谢之间的潜在联系。生成的数据似乎是可靠的,对数据的统计分析似乎很严格。讨论部分的大部分内容与研究结果一致。

小点:

1.第73行,T2DM缩写应预定义。

答: 非常感谢您的仔细审查,我们在介绍的第一段中首次出现的地方添加了T2DM的全名。

2.建议作者包括研究的目的/目标。

答:非常感谢您的建议,我们在修订稿的摘要中添加了研究的目的。

“本研究旨在探索miR-548ag调节葡萄糖代谢的分子机制。

再次感谢您的宝贵意见,这将使我们的摘要更加简洁和集中。

3.88-90行,请添加参考。

答: 非常感谢您的仔细审查。由于我们的疏忽,我们没有清楚地解释背景,对于给您对我们文章的理解造成的混乱,我们深表歉意。

关于miR-548ag生理功能的研究尚未在文献中报道。在我们的其他研究中,我们使用SNP芯片技术发现miR-548ag与1053个体的肥胖体征和生化指标呈正相关(图1 A-E)。此外,表达分析显示,与正常体重的个体相比,肥胖个体血清中miR-548ag的含量显着增加(图1 F)。以上结果尚未公开发表。

图1基因组关联分析,用于筛选和验证与肥胖发生相关的miRNA。 A HDL的相关miRNA。B LDL的相关miRNA。C TG的相关miRNA。D TC的相关miRNA。E FPG的相关miRNA。F 采集正常体重个体66例和肥胖个体116例血清标本,检测miR-548ag表达水平。*P<0.05.

4.结论部分缺失。

答:非常感谢您的仔细审查。根据您的建议,我们仔细阅读了《国际分子科学杂志》的投稿指南,并阅读了最近发表的文章。我们发现,该期刊并不要求将结论列为单独的部分,因此我们在文章末尾总结了结论。当然,如果需要明确的结论部分,我们会与编辑再次确认,如果需要,我们会立即修改。
